# COARSE-TUNING MODELS OF CODE WITH REINFORCEMENT LEARNING FEEDBACK

## ABSTRACT

Large Language Models (LLMs) pre-trained on code have recently emerged as the dominant approach to program synthesis. However, these models are trained using next-token prediction, which ignores the syntax and semantics of code. We propose RLCF, that further trains a pre-trained LLM via reinforcement learning, using feedback from a grounding function that scores the quality of the code. The grounding function uses (i) compiler-derived feedback on whether the code it generates passes a set of correctness checks; and (ii) feedback from a different LLM that compares the generated code to a reference code. RLCF is model- and language-agnostic. We empirically evaluate it on the MBJP and MathQA tasks for Java. Our experiments show that RLCF raises the odds that an LLM-generated program compiles, is executable, and produces the right output on tests, often allowing LLMs to match the performance of 2x-8x larger LLMs.

## 1 INTRODUCTION

Large language models (LLMs) pre-trained on code have had tremendous success in program synthesis in recent years (Chen et al., 2021a; Li et al., 2022; Austin et al., 2021). LLMs for code are typically trained to predict the next token in a program, given a sequence of prior tokens (Vaswani et al., 2017; Brown et al., 2020). However, supervised, next-token training overlooks two key aspects of computer programming. (i) There are many ways to solve the same programming problem, and multiple idiomatic expressions can lead to functionally equivalent codes. As such, there is no single correct answer to a programming problem. (ii) The semantics of code is well-modeled by a changing environment that tracks the set of variables in scope, their types and other properties, as well as the function being constructed over them, as a program is processed from start to finish. Programming constructs transform this environment, influencing the correct interpretation of subsequent code.

Thus, the process of writing code is perhaps better modeled as a Markov decision process (MDP), than as a next-token prediction problem. In the MDP, a positive reward is given when the LLM writes a code that matches the specification. An LLM can be trained to win the MDP via reinforcement learning (RL). Note that RL has been used for training LLMs to produce natural language, where humans judge the text to provide a reward (Stiennon et al., 2020; Ouyang et al., 2022).

Yet providing a reward in the case of code is challenging. Not only does it require detecting semantic and syntactic errors such as uninitialized variables, type discrepancies, non-terminating loops, etc., it also requires determining whether the code solves the programming problem. The most natural way to evaluate code to provide for a reward function is by executing the code on test cases (Le et al., 2022). While test cases offer an intuitive means to gauge code accuracy, having access to extensive unit test cases for a wide variety of real-life training codes is not realistic. If the goal is to power RL over a large, pre-existing code data set, another method of providing a reward is needed.

We present *Reinforcement Learning with Coordinated Feedback* (RLCF) as a way to train LLMs as a policy for writing code, using RL. Because this training happens after traditional pre-training but before task-specific fine-tuning, we call it *coarse-tuning*. In RLCF, the underlying MDP has a reward mechanism consisting of two parts:

1. First, after each action (token generation), a compiler is used to perform static analysis on the code generated so far. This analysis can detect simple syntactic errors, type errors, or even deeper semantic errors such as non-terminating loops. Detected static errors result in a negative reward.

2. Second, in the absence of test cases, it is important that the code still be a reasonable approximation to an example solution to the underlying programming task. Otherwise, the LLM could learn to "cheat" at avoiding static errors, for instance, by immediately terminating the program. However, because we are attempting to define a sequential game, we cannot simply compare the generated code to an example solution, as in supervised learning. Thus, we have a second LLM that has access to the example solution (without knowing that it is the example solution), and acts similar to a discriminator in imitation learning (Ho & Ermon, 2016). This LLM attempts to determine which program is being generated by the policy, and which is the example solution. If the policy can fool the discriminator, the policy receives a positive reward; otherwise, it is punished.

In the absence of test cases, both of these signals are vital to defining a useful reward function. The compiler alone cannot determine whether the generated code solves the programming problem at hand, though it can determine whether the generated program is syntactically and semantically valid. An LLM-based discriminator can determine whether the code as appropriate for the given problem, but has no formal understanding of the semantics of code and can easily be fooled by a policy that quickly learns to generate semantically incorrect codes. Together, these two components seem to provide an excellent reward signal. Our hypothesis is that a model coarse-tuned using RLCF will be better at downstream code generation tasks, compared to the same model that was pre-trained only using classical, next-token generation.

To evaluate this hypothesis, we implement RLCF on top of pre-trained CodeGen (Nijkamp et al., 2022), CodeT5 (Wang et al., 2021), and GPT-2 models (Radford et al., 2019). We evaluate the resulting models in the MBJP and MathQA programming tasks for Java (Athiwaratkun et al., 2022). Our experiments show that RLCF significantly raises the odds that an LLM-generated program compiles, is executable, and produces the right output on tests, and can allow LLMs to match the performance of $2\times$-$8\times$ larger LLMs (Black et al., 2021; Nijkamp et al., 2022). RLCF also significantly outperforms baselines based on supervised and reinforcement learning (Le et al., 2022) that also use compiler feedback.

To summarize, the contributions of this paper are as follows. First, we introduce the idea of coarse-tuning language models for code using RL feedback. Second, we introduce a way to balance the objectives of generating "natural" code and passing correctness checks by coordinating feedback from (i) a compiler and (ii) a discriminator LLM. Third, we implement RLCF and show that it significantly improves the ability of LLMs to generate code free of errors in downstream tasks.

## 2 RELATED WORK

**LLMs for Code Generation.** In recent years, significant research has been conducted in the field of formal-language generation. This research has primarily focused on the development of LLMs for code understanding and generation. These models have varying architectures, including Encoder-only (Kanade et al., 2020; Feng et al., 2020; Guo et al., 2020), Decoder-only (Chen et al., 2021a; Black et al., 2021; Nijkamp et al., 2022; Fried et al., 2022), and Encoder-Decoder transformers (Ahmad et al., 2021; Wang et al., 2021; Li et al., 2022). They have been pre-trained using self-supervised objectives, such as MLM (Devlin et al., 2018), RTD (Clark et al., 2020), DOBF (Roziere et al., 2021), Infilling (Fried et al., 2022; Bavarian et al., 2022), and Edit planning (Zhang et al., 2022), which aim to recover the original code from its obfuscated version. However, these de-noising objectives have been largely inspired by natural language and do not incorporate the syntactic and semantic rules that govern programming languages. As a result, these models are more likely to encounter syntax errors, type mismatches, and other runtime errors (Austin et al., 2021). Our proposed coarse-tuning routine is model agnostic and aligns better with these rules by using language-specific compiler feedback during auto-regressive code generation.

**Reinforcement Learning for LLMs for Natural Language.** Fine-tuning LLMs with human feedback (RLHF) has become the primary method for improving the quality of their natural-language responses (Stiennon et al., 2020; Ouyang et al., 2022; Ziegler et al., 2019). The central idea is to train them using a Preference Model that mimics human evaluation of response quality through a proxy reward function. Labelers are guided in real-time to compare the responses while ensuring researcher-labeler agreement. However, collecting large amounts of labeler annotations is expensive and challenging. Furthermore, this approach to collecting labels is not feasible for code due to its technical nature and complexity, often requiring domain experts with specialized knowledge and

a deep understanding of the context in which the code is used. The proposed RLCF is a coarse-tuning approach and does not require any human involvement by automating the feedback using a discriminator with access to widely available code-analysis tools.

**Semantics-Aware Program Synthesis.** Significant effort has been dedicated to capturing program semantics to guide synthesis. For example, (Chen et al., 2019; Ellis et al., 2019) conditions on the execution trace of generated code during the generation process. In contrast, Mukherjee et al. (2021) invokes a static analyzer to compute semantic attributes and direct further generation. Some researchers even approximate execution outcomes using neural models (Chen et al., 2021b; Inala et al., 2022; Le et al., 2022). However, these works demonstrate good performance in smaller domains, such as restricted Java, C, or toy programming languages like Karel (Pattis, 1994) and string processing (Gulwani, 2011), where aforementioned operations like execution tracing are inexpensive. In contrast, RLCF operates on full-scale Java or Python and uses static analysis to avoid executing generated programs. Execution can be challenging, requiring test cases and proper execution environment.

Another relevant work is CodeRL (Le et al., 2022), which also uses deep reinforcement learning in an actor-critic framework. Its critic is trained to evaluate the functional correctness of generated programs by executing them against unit tests during training. In practice, this limits the applicability of such proposals to specific types of fine-tuning tasks where whole, executable programs are generated and test cases are available. This limitation is a constraint on applicability. In contrast, application of RLCF only requires access to a static checker and a code corpus. As a result, it can be deployed when tests are not available, and used with any programming language that has static analyzers.

## 3 PROBLEM DEFINITION

We now define the problem of coarse-tuning a model of code with a contextual grounding function.

Assume a distribution of programming problems with solutions $F$, where sampling from $F$ produces a pair $(\mathbf{x}, \mathbf{y})$. Here, $\mathbf{x}$ is a *programming context* (for example, a natural-language prompt or a partially-finished program) and $\mathbf{y}$ is the response code (a sequence of tokens) associated with $\mathbf{x}$. In addition, we have access to a *grounding function* $G$, to recognize when a particular $\mathbf{y}$ is likely to have been sampled with a given $\mathbf{x}$. The function accepts two inputs: (i) a pair $(\mathbf{x}, \mathbf{y})$ sampled from $F$; and (ii) an alternative response code $\mathbf{y}'$. The grounding function attempts to determine whether the initial or the alternative response aligns with $\mathbf{x}$. In other words, $G$ identifies plausible $(\mathbf{x}, \mathbf{y})$ pairs. Importantly, while the programming context $\mathbf{x}$ is always the primary input to $G$, the function can take $\mathbf{y}$ and $\mathbf{y}'$ in any order thereafter. In an ideal scenario, a perfect grounding function, termed $G^*$ acts as an oracle. For any input where $(\mathbf{x}, \mathbf{y})$ was sampled from $F$, and $\mathbf{y}'$ is the distractor response code, $G^*(\mathbf{x}, \mathbf{y}, \mathbf{y}')$ returns 1. Conversely, $G^*(\mathbf{x}, \mathbf{y}', \mathbf{y})$ results in -1, indicating $(\mathbf{x}, \mathbf{y})$ is more plausible than $(\mathbf{x}, \mathbf{y}')$.

The problem of *coarse-tuning* is as follows. Given a pre-trained code model $f_\theta(\mathbf{y}|\mathbf{x})$ parameterized on $\theta$, an appropriate "anchor" loss function $L_{\text{anc}}$, and a grounding function $G$, choose $\theta'$ to minimize:

$$\mathbb{E}_{(\mathbf{x},\mathbf{y})\sim F}\left[L_{\text{anc}}(\theta'|\mathbf{x},\mathbf{y},\theta) + \alpha\mathbb{E}_{(\mathbf{y}'\sim f_{\theta'}(.|\mathbf{x}))}\left[G(\mathbf{x},\mathbf{y},\mathbf{y}')\right]\right] \tag{1}$$

**Discussion.** Intuitively, we are attempting to choose a parameter set $\theta'$ for our model that "fools" the grounding function on programming problems sampled from $F$. That is, $G$ will not be able to determine whether (on the one hand) a response sampled from our learned model $f_{\theta'}(.|\mathbf{x})$ is the "real" response, or (on the other hand) the actual response is the "real" one. Note that each generated $\mathbf{y}$ is treated holistically by $G$, and not merely as a sequence of tokens.

The purpose of the anchor loss function is to ensure that the newly-learned parameter set $\theta'$ does not differ too much from the original parameter set $\theta$. Many anchors are possible. For example, assuming an LLM that produces tokens sequentially, the anchor loss could measure the divergence between $f_{\theta'}(.|\mathbf{x})$ and $f_\theta(.|\mathbf{x})$ computed at each token in $\mathbf{y}$. We will consider this particular anchor loss subsequently in the paper.

## 4 DESIGN OF THE GROUNDING FUNCTION

**Both Static Semantic Analysis and an LLM Are Necessary.** As intimated in the introduction of the paper, our grounding function makes use of both a static semantic analysis (essentially, a

```
1  // Starter Code
2  import java.io.*;
3  import java.util.*;
4  class Main{
5  public static void main(String[] args){
6      /*<your_code_here>*/
7  }
8
9  // Ground truth response
10 Scanner sc=new Scanner(System.in);
11 int a=sc.nextInt(),b=sc.nextInt(), c=sc
       .nextInt();
12 System.out.println((a+b+c)>21? "bust":"
       win")
13 }
14
15 // RLCF response
16 Scanner sc = new Scanner(System.in);
17 int a = sc.nextInt();
18 int b = sc.nextInt();
19 int c = sc.nextInt();
20 if(a+b+c>=22) {
21     System.out.println("bust");
22 }else {
23     System.out.println("win");}
24 }
```

```
1
2  // Iteration 25k
3  String s=null;
4  Scanner sc=n.readLine().split(" ");
5  int a=Integer.parseInt(sc.next());
6  int b=Integer.parseInt(sc.next());
7  int c=Integer.parseInt(sc.next());
8  if(a+b+c>=22){s= "bust";}
9  if(s===="win"){s= "bust";}
10 System.out.println(s);
11 }
12
13 // Iteration 50k
14 Scanner<String> stra =! (String.valueOf(stra.
       charAt(0)) + String.valueOf(stra.charAt
       (1)) + String.valueOf(stra.charAt(2))
       >=22? "bust":"win");
15 System.out.println(stra.charAt(0) + " "+stra.
       charAt(1) + " "+stra.charAt(2));
16 }
17
18 // Iteration 75k
19 System.out.println(Kojiscashin.compareTo(Kojis
       cashin.compareTo(Kojiscashin.next)) >=
       0? "bust" : "win");
20 }
```

```
1  // Iteration 25k
2  Scanner sc = new Scanner(System.in);
3  int a = sc.nextInt();
4  int b = sc.nextInt();
5  int c = sc.nextInt();
6  if(a+b+c >= 22){a=7;}
7  if(a+b+c >= 21){a=8;}
8  if(a+b+c >= 21){b=8;}
9  if(a+b+c <= 21){c=8;}
10 if(a+b+c <= 21){c=8;}}
11
12 // Iteration 75k
13 Scanner sc = new Scanner(System.in);
14 int a = sc.nextInt();
15 int b = sc.nextInt();
16 int c = sc.nextInt();
17 int sum = a+b+c;
18 if(sum >= 13){
19     System.out.println("bust");
20     return;}
21 if(sum >= 21){
22     System.out.println("win");
23     return;}
24 if(sum >= 22){
25     System.out.println("bust");
26     return;}}
```

| (a) RLCF | (b) RLCF w Discriminator only | (c) RLCF w Compiler only |

Figure 1: `main` method completed by CODET5+RLCF for the prompt: *"Given three integers, if their sum is >= 22, print bust; else, print win."* grounded with (a) discriminator and compiler. (b) discriminator only: output generated during training closely resembles the reference but starts to show trivial errors (highlighted in red). (c) compiler only: output passes all static checks but is nonsensical.

compiler) as well as an LLM. We will describe how it does this formally in a moment, but at a high level: given arguments $(\mathbf{x}, \mathbf{y}_0, \mathbf{y}_1)$, the grounding function $G$ always first attempts to compile both $\mathbf{y}_0$ and $\mathbf{y}_1$. If one compiles and the other does not, it returns immediately with a 1 or a $-1$, as described in Section 3. However, if both compile, the function then invokes the LLM with parameter set $\omega$, $D_\omega(\mathbf{x}, \mathbf{y}_0, \mathbf{y}_1)$ to deduce its return value. This *discriminator* LLM is a specifically trained neural network designed to distinguish between (prompt, response) pairs originating from $F$, and responses generated by $f_\theta$.

One may ask: Why not solely rely on the compiler or just the discriminator for the grounding function? The problem with relying solely on a discriminator is that it does not have an unassailable understanding of code syntax and static semantics, such as type correctness. Without the compiler, a well-trained policy will find weaknesses in the discriminator, and begin to generate codes that "look" correct to the discriminator, but have obvious errors. And while the compiler can never be fooled into accepting a code with incorrect syntax or a type error, it cannot guarantee that a generated code is responsive to its context. The result is that if a LLM is coarse-tuned using a grounding function that has only one of the two, the generated programs are quite terrible. Refer to Figure 1.

**Using compiler for grounding.** The compiler is a Boolean function $C(\mathbf{x}, \mathbf{y})$ that evaluates to TRUE if $y$ compiles, FALSE otherwise. Regardless of whether or not the grounding function $G$ is perfect, it always uses the compiler correctly such that:

$$\text{ROLLOUTTRAJECTORY } (\mathbf{x}, \mathbf{y}, \omega, \theta')$$

$t \leftarrow 0$
**repeat**
    $t \leftarrow t + 1$
    $y'_t \sim f_{\theta'}(.|y'_1, ..., y'_{t-1}, \mathbf{x})$
**until** $y'_t = \text{EOP}$
**if** $\mathbf{y}'$ compiles **then**
    **return** $(\mathbf{y}', D_\omega(\mathbf{x}, \mathbf{y}', \mathbf{y}))$
**else**
    Determine $j$, the location at
        which compilation failed
    **return** $(\langle y'_1, ..., y'_j \rangle, -1)$
**end if**

Figure 2: Producing a (reward, trajectory) pair in RLCF.

- If $C(\mathbf{x}, \mathbf{y}_0) \wedge \neg C(\mathbf{x}, \mathbf{y}_1)$, then $G(\mathbf{x}, \mathbf{y}_0, \mathbf{y}_1) = 1$.
- If $\neg C(\mathbf{x}, \mathbf{y}_0) \wedge C(\mathbf{x}, \mathbf{y}_1)$, then $G(\mathbf{x}, \mathbf{y}_0, \mathbf{y}_1) = -1$.

Because all programs sampled from $F$ compile, and the grounding function always uses $C$ to determine the "real" response, coarse-tuning with RLCF will produce a model that is penalized if it does not produce a compilable program. However, we can do better in the case that a $-1$ returned by the grounding function is due to the compiler not accepting the generated program. Virtually any

compiler will be able to provide some sort of localization of the source of an error, and using that information during training will help to isolate the particular decision or decisions in the generation of $\mathbf{y}'$ that led to the compiler failure.

For example, imagine that we have generated the line of Java code "`content.append((char) value);`" and yet the type of the variable `content` does not have a method `append`. A Java compiler will likely point to the invocation of `append` as the source of the error. This localization is not perfect; is the problem that the code the model has chosen invokes the wrong method, or is the problem that the model has chosen the correct method, but the wrong type for `content`? Still, a compiler error at the invocation of `append` almost assuredly means that the issue in the generated code occurs at or before the call on `append`, and so we truncate the generated sequence at the call and give a $-1$ reward at that point. Figure 2 gives pseudo-code for the algorithm that produces a (trajectory, reward) pair based on the foregoing idea. As most compilers are not capable of processing an input incrementally, we first generate an entire response $\mathbf{y}'$, and then truncate the code after the first compilation error.

**Design of the discriminator.** Our particular implementation of $D$ makes use of a CodeBERT (Feng et al., 2020) network used to produce embedding(s) for $\mathbf{y}_0$ and $\mathbf{y}_1$. Each embedding is then fed into a multi-layer perceptron (MLP) that processes the embedding into a single number, representing a level of belief or certainty that a response was produced by $F$, as opposed to the code model $f$. The more certain the model is that the response was produced by $F$, the larger the output of the MLP. A $\tanh$ over the difference then produces the final score:

$$D_\omega(\mathbf{x}, \mathbf{y}_0, \mathbf{y}_1) \equiv \tanh\left(\text{MLP}\left(\text{CodeBERT}\left(\mathbf{x} \circ \mathbf{y}_0\right)\right) - \text{MLP}\left(\text{CodeBERT}\left(\mathbf{x} \circ \mathbf{y}_1\right)\right)\right)$$

Here, $\circ$ is the tensor concatenation operator. $\omega$ is the parameter set for MLP and CodeBERT model. Because both of the CodeBERT models share the same parameterization (likewise the two MLPs), it is not possible for the model described above to learn any sort of positional preference. That is, it cannot learn that the first candidate response is more likely to come from $F$, for example. Hence, to train $D_\omega$, we need not worry about the parameter ordering, and it suffices to choose $\omega$ so as to minimize the following expectation:

$$\mathbb{E}_{(\mathbf{x},\mathbf{y}) \sim F}\left[\mathbb{E}_{(\mathbf{y}' \sim f_\theta(.|\mathbf{x}))}\left[D_\omega(\mathbf{x}, \mathbf{y}', \mathbf{y})\right]\right] \tag{2}$$

Note that this minimization must be done with respect to a particular parameterization of the model $f$. The above expression defines the minimization as happening with respect to $\theta$, the original parameterization before we undertake coarse-tuning with compiler feedback. However, as we discuss in the next section, we may continue to update the parameter set $\omega$ by minimizing with respect to $\theta'$ during reinforcement learning, at the same time that we learn $\theta'$.

## 5 REINFORCEMENT LEARNING

Now given a particular anchor loss and grounding function, one could imagine using a gradient-based method to learn $\theta'$. At each iteration, sample $(\mathbf{x}, \mathbf{y})$ from $F$. First, compute the gradient of $L_{\text{anc}}$ with respect to $\theta'$; next, compute the gradient of $\mathbb{E}_{(\mathbf{y}' \sim f_{\theta'}(.|\mathbf{x}))}\left[G(\mathbf{x}, \mathbf{y}, \mathbf{y}')\right]$ with respect to $\theta'$; combine the two and take a step in the direction of steepest descent. However, there are two obvious problems with this approach. First, $f$ is an LLM and $\mathbf{y}' = \langle y_1', y_2', ..., y_T' \rangle$ is a sequence produced via recursive generation of tokens. Differentiating an expectation of a function computed over a sequence of tokens generated by an LLM is challenging, to say the least. Second, while $G$ may use a neural network that is differentiable, it also uses a compiler, which almost assuredly is not. Thus, we resort to reinforcement learning to deal with the inner expectation in Eqn. (1).

Specifically, program synthesis can be modeled as a Markov decision process (Chen et al., 2020). In this framework, a program is recursively sampled as $y_t' \sim f_{\theta'}(.|\mathbf{x}, \mathbf{y}_{<t}')$ until either an "End-Of-Program" token is generated or the maximum decoding length (the MDP's horizon) is reached. We reward the sequence of decisions embodied by $\mathbf{y}'$ using $G(\mathbf{x}, \mathbf{y}', \mathbf{y})$. A detailed explanation of this MDP-based formulation of program synthesis can be found in Appendix 8.3.

We present our final algorithm as Algorithm 1 (See Appendix section 8.1). RLCF uses Generalised Advantage Estimation (GAE) (Schulman et al., 2015) along with Proximal Policy Optimization (PPO; a type of actor-critic RL) (Schulman et al., 2017) to perform reinforcement learning. PPO requires

| MODEL | PASS@K | | | EXEC@K | | | COMP@K | | |
|---|---|---|---|---|---|---|---|---|---|
| | K=1 | K=10 | K=100 | K=1 | K=10 | K=100 | K=1 | K=10 | K=100 |
| CODEGEN (350M) | 4.54 | 8.64 | 11.34 | 54.63 | 80.93 | 89.69 | 60.68 | 86.18 | **94.32** |
| + RLCF | **5.08** | **10.15** | **13.40** | **63.58** | **86.11** | **91.23** | **71.82** | **90.45** | **94.32** |
| CODET5 (770M) | 4.17 | 7.66 | 9.27 | 48.85 | 79.52 | 88.65 | 56.44 | 84.91 | 91.23 |
| + RLCF | **6.32** | **12.26** | **16.49** | **54.09** | **82.52** | **89.69** | **62.68** | **87.56** | **92.78** |
| GPT2 (1.5B) | 1.88 | 5.28 | 7.21 | 39.35 | 62.83 | 72.68 | 44.42 | 66.86 | 74.22 |
| + RLCF | **2.03** | **6.56** | **10.82** | **44.58** | **79.42** | **87.62** | **53.10** | **83.28** | **89.17** |

Figure 3: Metrics tracking functional correctness and error profiles on MBJP.

one more neural network, the critic network $v_\varphi$. This network is similar to the LLM $f_\theta$, except that the language modeling head is replaced with an MLP used to perform value estimation. In lines 3 and 4, we first bootstrap the policy using Supervised Learning and pre-train the discriminator. Finally, for $N$ episodes, we train our policy and the critic by maximising the expected discounted return using PPO. Note that for a generated response $\mathbf{y}'$, a portion of the reward associated with token $y'_{|\mathbf{y}'|}$ (the last token) is generated by the grounding function. However, each token is also associated with a portion of the anchor loss, which for learned parameter set $\theta'$ and initial parameter set $\theta$ is $\beta \log(\frac{f_{\theta'}(y'_j|\mathbf{x},\mathbf{y}'_{<j})}{f_\theta(y'_j|\mathbf{x},\mathbf{y}'_{<j})})$. This quantity is subtracted from the reward associated with each token. Also note that if the input parameter "freezeDisc" is `false`, then during reinforcement learning the discriminator network is continuously trained to counter the updated policy in an adversarial fashion.

# 6 EXPERIMENTS

## 6.1 EFFECTIVENESS OF RLCF AS A COARSE-TUNING APPROACH

The central hypothesis of the paper is that RLCF can be used to make any LLM-based model for code more useful for downstream tasks.

To test this hypothesis, we choose three different pre-trained LLMs, do coarse-tuning using RLCF, and then test whether the resulting models are more accurate for programming tasks than the original models without coarse-tuning. Specifically, the three LLMs we consider are (i) the 350M parameter CodeGen-multi model (Nijkamp et al., 2022) pre-trained on PILE (Gao et al., 2020) and BigQuery (Google); (ii) the 770M parameter encoder-decoder CodeT5 model (Wang et al., 2021) pre-trained on Code-SearchNet (Husain et al., 2019); and (iii) the 1.5B parameter decoder-only GPT-2 model (Radford et al., 2019) pre-trained using NTP on WebText (Radford et al., 2019). For each of them, we do coarse-tuning using RLCF over programs sampled from CODENETJAVA (Puri et al., 2021), which consists of 75 thousand programs solving approximately 250 problems in Java. To produce a (prompt, response) pair, we randomly pick a method in the code. The code up to that point is the prompt and response is the remainder method body. Refer to Appendix §8.10 for illustration of data preparation.

We selected Java as the prototype for RLCF due to its static typing, which facilitates precise compiler feedback on compile-time issues. Moreover, Java boasts a rich ecosystem of established static analyzers, such as ErrorProne[1], offering advanced static checks. Beyond these checks, we detect and penalize unused variables in the code using the tree-sitter library for Java[2], equally penalising them at their declarations.

To evaluate the utility of all models, we use the Mostly Basic Java Problems (MBJP) (Athiwaratkun et al., 2022). MBJP consists of 966 programming problems, each containing a short natural-language prompt, a canonical solution in Java, and 3 corresponding tests. We first fine-tune each model using MBJP, then test. We use a 80-20 testing/fine-tuning split. Refer to Appendix §8.4 for all the implementation details.

Results are given in Figure 3. "+ **RLCF**" refers to RLCF with adversarial training. "PASS@$k$" refers to the fraction of test problems for which, if $k$ responses are generated using the LLM, there exists at

---

[1]https://errorprone.info/

[2]https://github.com/tree-sitter/tree-sitter-java

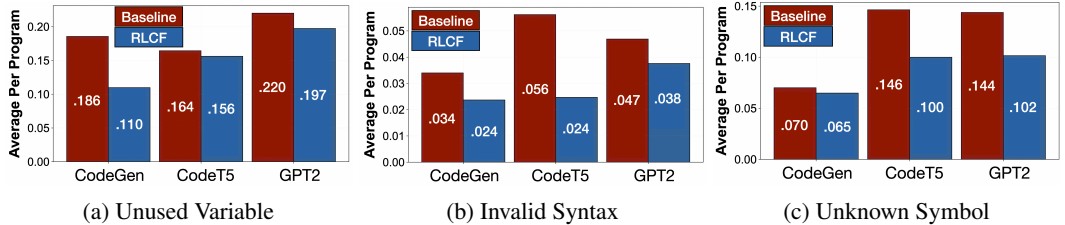

|                | (a) Unused Variable | (b) Invalid Syntax | (c) Unknown Symbol |

Figure 4: Prevalence of model exhibited bad practices (a) and various compilation errors (b), (c) on the MBJP test data set. Each plot shows their average number observed per generated program.

least one response that compiles, executes all tests without crashing, and passes the tests. "EXEC@$k$" refers to fraction of test problems where there exists at least one response that compiles and executes all tests without crashing. "COMP@$k$" refers to the fraction where at least one response compiles.

We find that in every case, coarse-tuning with RLCF results in a model that can produce codes that are more likely to compile, to execute if they compile, and to pass the test cases if they compile and execute. Considering the increased ability to pass all test cases, the increase is either moderate (consider CODEGEN with $k = 100$, where there is an increase from 11.3% to 13.4%) to significant (consider CODET5 with $k = 100$, where there is an increase from 9.3% to 16.5%). Considering that RLCF is a very lightweight way to perform coarse-tuning (we performed RLCF on only 75K Java programs) these improvements seem significant. The boost in test-case pass rate is notable, especially as RLCF trains without direct access to test-cases. Instead, it relies on a discriminator to stay on-target. By allowing the policy to go beyond training programs and receive feedback from the grounding function, it evidently enhances the model's grasp of natural language prompts, leading to more accurate results in MBJP.

In Figure 11, we show the prevalence of various compilation error categories on the MBJP data set before and after coarse-tuning with RLCF, and find that indeed, the incidence of various compilation errors decreases. This trend is observed across all three models tested.

We repeated the same experiment with a second data set: MathQA for Java (Athiwaratkun et al., 2022). It has 1883 problems, which require "understanding" what the English prompt is asking for, but have simple arithmetic solutions. This data set is an interesting test case because the difficulty is not writing the code; rather, it is dealing with the prompt. Thus, we speculate that RLCF may be less useful. We do a 50-50 train-test split and fine-tune each model before testing. We show the PASS@$k$ results in Figure 5. Because solutions for MathQA are simple, compilation and execution are easy, thus we focus only on the pass rate. Even in this case, where generating a code that compiles is easy, we still find a modest increase (consider CODET5 with $k = 100$, where there is an increase from 31.7% to 32.2%) to a significant increase (consider GPT2 with $k = 10$, where there is an increase from 13.4% to 19.0%). This further confirms that with RLCF, models better understand the prompt.

| MODEL | K=1 | K=10 | K=100 |
| --- | --- | --- | --- |
| CODEGEN (350M) | 18.21 | 23.59 | 28.02 |
| + RLCF | **18.82** | **24.34** | **29.93** |
| CODET5 (770M) | 10.28 | 20.19 | 31.63 |
| + RLCF | **12.69** | **21.63** | **32.16** |
| GPT2 (1.5B) | 3.26 | 13.36 | 33.01 |
| + RLCF | **7.71** | **19.02** | **34.07** |

Figure 5: PASS@K performance comparison on 942 problem MathQA-Test

A secondary hypothesis of the paper was that there would be an increase in utility associated with RLCF, beyond having a larger number of generated programs that would compile. Intuitively, perhaps giving a LLM access to a grounding function will make the LLM a better programmer, teaching it basic rules about programming, and hence more able to produce useful codes, regardless of compilation. These results seem to validate this hypothesis. For MathQA, almost all generated programs compile, so increases in test-case pass rate cannot be attributed to increased compilation rate. We also see this in MBJP. Consider CODET5 with $k = 100$. There is a modest 1.5% increase in compilation rate, but a much more significant increase in test-case pass rate. Thus, there are auxiliary benefits to RLCF, beyond increasing compilation rate.

| MODEL | PASS@K | | | EXEC@K | | | COMP@K | | |
|---|---|---|---|---|---|---|---|---|---|
| | K=1 | K=10 | K=100 | K=1 | K=10 | K=100 | K=1 | K=10 | K=100 |
| CODEGEN (2.8B) | 6.42 | 13.15 | 18.04 | 63.87 | 86.79 | 92.26 | 72.74 | 89.97 | 93.29 |
| GPT-NEO (1.3B) | 3.43 | 9.68 | 14.94 | 51.60 | 82.56 | 91.75 | 60.56 | 87.47 | 93.81 |
| **RLCF** (770M) | 6.32 | 12.26 | 16.49 | 54.09 | 82.52 | 89.69 | 62.68 | 87.56 | 92.78 |
| **RLCF** (350M) | 5.08 | 10.15 | 13.40 | 63.58 | 86.11 | 91.23 | 71.82 | 90.45 | 94.32 |

Figure 6: Comparison with larger LLMs on MBJP.

| MODEL | PASS@K | | | EXEC@K | | | COMP@K | | |
|---|---|---|---|---|---|---|---|---|---|
| | K=1 | K=10 | K=100 | K=1 | K=10 | K=100 | K=1 | K=10 | K=100 |
| CODEGEN (350M) | 4.54 | 8.64 | 11.34 | 54.63 | 80.93 | 89.69 | 60.68 | 86.18 | 94.32 |
| + MONO | **5.20** | 8.87 | 11.34 | 56.55 | 82.96 | 90.72 | 66.18 | 89.01 | 93.29 |
| + BIPOLARRAMP | 5.03 | 9.07 | 12.88 | 58.42 | 81.33 | 86.59 | 66.83 | 86.51 | 89.69 |
| + CODERL | 3.62 | 6.35 | 9.28 | 62.33 | 82.57 | 88.14 | **72.39** | 88.99 | 93.30 |
| **+ RLCF W FIXDISC** | 5.04 | 10.06 | **13.91** | 61.47 | 85.71 | **93.81** | 70.02 | 89.91 | **95.36** |
| **+ RLCF** | 5.08 | **10.15** | 13.40 | **63.58** | **86.11** | 91.23 | 71.82 | **90.45** | 94.32 |

Figure 7: Ablation study with CodeGen (350M) on MBJP.

## 6.2 COMPARISON WITH LARGER LANGUAGE MODELS

One benefit of RLCF is that coarse-tuning can result in a model that performs on par with a larger (and hence more costly) model—without the associated difficulty of dealing with a large model. In Figure 6, we compare our approach's smaller models CodeGen (350M) and CodeT5 (770M) with 2x-8x times larger models: GPT-Neo (1.3B) (Black et al., 2021) and CodeGen-multi (2.8B) (Nijkamp et al., 2022). We observe that CodeGen with RLCF, at only 350M parameters, performs almost on par with GPT-Neo (which is 4x larger). CodeGen with RLCF is slightly better for PASS@1 and PASS@10, but slightly worse for PASS@100. We observe that CodeT5 with RLCF at 770M parameters, performs almost on par with the larger (and quite high-quality) CodeGen model, which is $3.5\times$ larger. This result shows that with proper tuning, code-generation models can be improved to a significant-enough extent that it makes up for a deficiency in model size.

## 6.3 ABLATION STUDIES

RLCF embodies several different ideas: doing coarse-tuning with language-specific training, the use of a compiler to assign "blame" for errors in the code, and the use of reinforcement learning. We present an ablation study aimed at determining which parts of the overall RLCF are actually instrumental in achieving the performance gains. Using the MBJP data set, for both the CODEGEN and CODET5 models, we consider the following options.

| MODEL | K=1 | K=10 | K=100 |
|---|---|---|---|
| CODEGEN (350M) | | | |
| + RLCF W FIXDISC | 17.78 | **24.34** | 29.40 |
| **+ RLCF** | **18.82** | 24.34 | **29.93** |
| CODET5 (770M) | | | |
| + RLCF W FIXDISC | 11.62 | 21.17 | 32.05 |
| **+ RLCF** | **12.69** | **21.63** | **32.16** |

Figure 9: Ablation tests (with and without adversarially-trained discriminator) on MathQA. PASS@$k$ rates are shown.

**(1) Ablation with Mono**: Nijkamp et al. (2022) observes that monolingual supervised training of multilingual models boosts performance in downstream tasks in the target language. Could it be the case that our gains are due primarily to the fact that RLCF is simply finishing training with Java, as opposed to its use of reinforcement learning? To answer this, we coarse-tuned with standard cross entropy loss to obtain a fully supervised baseline (formulated in L3 in Algorithm 1), rather than using RLCF.

**(2) Ablation with Bipolar Ramp**. Is reinforcement learning needed? How well does RLCF perform compared to a supervised baseline that uses compiler feedback? No prior work attempts this, so we devised a weakly supervised baseline trained to minimise token-wise Bipolar RAMP loss

| MODEL | PASS@K | | | EXEC@K | | | COMP@K | | |
|---|---|---|---|---|---|---|---|---|---|
| | K=1 | K=10 | K=100 | K=1 | K=10 | K=100 | K=1 | K=10 | K=100 |
| CODET5 (770M) | 4.17 | 7.66 | 9.27 | 48.85 | 79.52 | 88.65 | 56.44 | 84.91 | 91.23 |
| + MONO | 4.68 | 9.10 | 11.85 | 50.77 | 78.95 | 88.65 | 59.90 | 87.28 | 92.78 |
| + BIPOLARRAMP | 5.93 | 10.43 | 12.88 | 54.73 | 80.47 | 87.11 | **63.90** | 86.82 | 92.78 |
| + CODERL | 4.15 | 9.47 | 12.88 | 50.72 | 81.21 | 90.20 | 61.65 | **88.09** | 93.29 |
| + RLCF W FIXDISC | **6.60** | **12.71** | **16.49** | **54.71** | 81.46 | **91.23** | 63.62 | 87.60 | **93.81** |
| + RLCF | 6.32 | 12.26 | **16.49** | 54.09 | **82.52** | 89.69 | 62.68 | 87.56 | 92.78 |

Figure 8: Ablation study with CodeT5 (770M) on MBJP.

where tokens associated with failed compilation are penalized (Jehl et al., 2019). Thus, supervised learning is used to localize the error associated with failed compilation, as opposed to RL.

**(3) Ablation with CodeRL**. How does RLCF stack up against established reinforcement learning methods? We tested this by training CodeRL, a known model from Le et al. (2022), without unit-tests, where rewards were given based solely on whether the code compiled successfully or not.

**(4) Ablation with fixed discriminator**. How necessary is adversarial training? Perhaps it is possible to achieve the same results by fixing the discriminator after pre-training. To test this hypothesis, we run RLCF with freezeDisc set to `true`.

The results (Figures 7 and 8) seem to show that there is a significant advantage of RLCF compared to simply finishing training with Java (*Mono*), or training using just compiler feedback with supervised learning(*BipolarRamp*) or reinforcement learning (*CodeRL*). In nearly every case, either RLCF (with the fixed discriminator, or with an adversarial setting) was the best approach. What is less clear is whether or not adversarial training was actually advantageous. Figures 7 and 8 did not seem to show a significant difference between the two methods, with the fixed discriminator winning more often, though not by much. We also show ablation results comparing RLCF with and without a fixed discriminator in Figure 9. This figure shows that for MathQA, RLCF with adversarial training typically wins, though the difference in PASS@$k$ rates appears insignificant. Given all of this, both options seem equivalent. It could be that the anchor loss we use keeps the learned model parameter set $\theta'$ relatively close to the initial set, and so updates to the discriminator are not necessary. It may also be that adversarial training—most commonly associated with GANs (Ho & Ermon, 2016)—is simply challenging to use.

# 7 CONCLUSIONS AND FUTURE WORK

Our research findings demonstrate that our novel coarse-tuning method leads to significantly improved code generation in LLMs. Specifically, our approach results in improved functional correctness and a lower likelihood of errors in the generated code, leading to proper compilation and successful execution of the code. To the best of our knowledge, other researchers have not taken this approach in their studies of code generation in LLMs, making our approach both novel and unique.

**Limitations:** During policy training, analyzing the code can take a lot of time, which can be a bottleneck. To sidestep this issue, we trained our model using CODENETJAVA (Puri et al., 2021), which only contains standalone programs, with no dependencies on user-defined packages or libraries. However, we would like to use larger publicly available dataset(s) that do have these dependencies. To make this possible, we need a faster method of static analysis.

**Broader Impact:** Our contribution to new knowledge is the development of a novel approach to code generation with LLMs that models it as a Markov decision process with immediate feedback guided by static analysis and code correctness via a grounding function. The significance of our work lies in its potential to contribute to the development of more accurate and efficient LLMs for code generation, which could have a significant impact on software engineering and related fields.

**Future Work:** In terms of future research, our work demands and paves the way for deeper semantic analysis of the generated code, to further improve the performance of LLMs on code-generation tasks. Furthermore, our work questions the conventional approach of treating code as natural language.

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

# 8 APPENDIX

## TABLE OF CONTENTS

## 8.1 ALGORITHM OUTLINE

---

**Algorithm 1** RLCF: Reinforcement Learning with Coordinated Feedback

---

1: **Input:** $\{\theta, \gamma, \lambda, \epsilon, \text{freezeDisc}\}$ // $\theta$ are the pre-trained LLM weights, $\lambda$ the GAE discount, $\gamma$ the RL discount, $\epsilon$ the PPO clip parameter, freezeDisc is `false` if we perform adversarial learning

2: **Output:** $\theta'$

3: **initialize the policy.** Starting with initial parameter set $\theta$, perform gradient descent to update $\theta$ so as to minimize $\mathbb{E}_{(\mathbf{x},\mathbf{y}) \sim F}\left[ \sum_i^T - \log f_\theta(y_i|\mathbf{x}, \mathbf{y}_{<i}) \right]$

4: **initialize the discriminator.** Trainable discriminator parameter set $\omega$ includes the Code-BERT model weights and the MLP weights. Starting with pre-trained CodeBERT weights and a randomly-initialized MLP, perform gradient descent on $\omega$ so as to minimize $\mathbb{E}_{(\mathbf{x},\mathbf{y}) \sim F}\left[ \mathbb{E}_{(\mathbf{y}' \sim f_\theta(.|\mathbf{x}))}\left[ D_\omega(\mathbf{x}, \mathbf{y}', \mathbf{y}) \right] \right]$

5: **initialize the critic.** The language-model portion of parameter set $\varphi$ for critic $v_\varphi$ is set to $\theta$. The MLP head $v$ is initialized randomly

6: $\theta'^{(1)} \leftarrow \theta; \omega^{(1)} \leftarrow \omega; \varphi^{(1)} \leftarrow \varphi$

7: **for** $i = 1$ **to** $N$ **do**

8:     Sample an $(\mathbf{x}, \mathbf{y})$ pair from $F$

9:     $(\mathbf{y}', R) \leftarrow \text{ROLLOUTTRAJECTORY}(\mathbf{x}, \mathbf{y}, \omega^{(i)}, \theta'^{(i)})$

10:     **for each** $j \in \{1, ..., |\mathbf{y}'|\}$, $r_j \leftarrow -\beta \log(\frac{f_{\theta'^{(i)}}(y_j'|\mathbf{x},\mathbf{y}_{<j}')}{f_\theta(y_j'|\mathbf{x},\mathbf{y}_{<j}')})$ // Add in the anchor loss at each step

11:     $r_{|\mathbf{y}'|}$ += $R$ // Add in the final discriminator reward

12:     **if not** (freezeDisc) **then**

13:         **update the discriminator.** Use sampled $(\mathbf{x}, \mathbf{y})$ and $\mathbf{y}'$ to perform a step of gradient descent from $\omega^{(i)}$ to choose $\omega^{(i+1)}$ to minimize $D_{\omega^{(i+1)}}(\mathbf{x}, \mathbf{y}', \mathbf{y})$

14:     **else**

15:         $\omega^{(i+1)} \leftarrow \omega^{(i)}$

16:     **end if**

17:     $\{(G_j, A_j)\}_{j=1}^{|\mathbf{y}'|} \leftarrow \text{GAE}_{\gamma,\lambda}\left(\{r_j\}, \mathbf{x}, \mathbf{y}', \varphi^{(i)}\right)$ // Estimate returns $G$ and advantages $A$

18:     **for each** $j \in \{1...|\mathbf{y}'|\}$, define the function $\rho_j(\theta'^{(i+1)}) = \frac{f_{\theta'^{(i+1)}}(y_j'|\mathbf{x},\mathbf{y}_{<j}')}{f_{\theta'^{(i)}}(y_j'|\mathbf{x},\mathbf{y}_{<t}')}$

19:     **update the policy.** Use gradient ascent to choose $\theta'^{(i+1)}$ to maximize
$\sum_j \min\left( \rho_j(\theta'^{(i+1)})A_j, \quad \text{clip}(\rho_j(\theta'^{(i+1)}), 1 - \epsilon, 1 + \epsilon)A_j \right)$

20:     **update the critic.** Use gradient descent to choose $\varphi^{(i+1)}$ to minimize
$\sum_j \left( G_j - v_{\varphi^{(i+1)}}(\mathbf{x}, \mathbf{y}_{<j}') \right)^2$

21: **end for**

22: $\theta' \leftarrow \theta'^{(N+1)}$

23: **return** $\theta'$

---

## 8.2 ILLUSTRATION: PRE-TRAINING, COARSE-TUNING AND FINE-TUNING

Figure 10: Block Diagram showing schematic of proposed framework

The architectural nuances of our framework are illustrated in Fig. 10. We've partitioned our framework into three progressive stages:

1. **Initialization of Policy and Discriminator**
   - LLM undergoes a few initial training epochs on CODENETJAVA (Puri et al., 2021) using supervised learning for bootstrapping the policy.
   - CodeBERT, a known model, is pre-trained as described in Section 8.4.2. It's then combined with a randomly-initialized MLP-head to initiate the discriminator $D$.
   - Before doing RLCF, we pre-train $D$ based on responses sampled from the initialized policy. Refer to Section 8.8 for an in-depth discussion.

2. **Coarse-tuning of Policy with RLCF**
   - LLM's next stage of training is 'coarse-tuning'. It undergoes this either in an adversarial setting or with a static discriminator (when *freezeDisc* is set to True). Refer Algorithm 1 for specifics.

3. **Fine-tuning Stage**
   - LLM gets refined for downstream tasks using supervised learning on selected evaluation datasets.

**Inference.** Finally, during inference, each LLM is given a prompt for a problem from the test-split of MBJP/MathQA, which includes a natural language question and starter code to complete. Given a $k$ value, model generates $k$ potential solutions. These solutions are tested against provided unit-tests. Three metrics, comp@k, exec@k, and pass@k, measure the success of these solutions on criteria of compilation, execution, and full test-case pass rates. To understand the unbiased metric estimation, see Section 8.4.5.

### 8.3 MDP FOR CODE GENERATION

MDP can be represented using the tuple $(\rho, \mathcal{S}, \mathcal{A}, P, \mathcal{R}, \gamma, T)$, where $\rho$ is the distribution over initial states, $s_o$ comprised of the prompts provided for code generation. $\mathcal{S}$ is the program state space and $\mathcal{A}$ is the action space formed by the vocabulary of program tokens.

Initially, the MDP starts with $s_o = \mathbf{x}$ where the prompt $\mathbf{x}$ along with response code $\mathbf{y}$ is sampled from $F$. At each time-step of the MDP, policy $f_\theta$ takes an action $a_t$ by predicting the next token $y'_{t+1}$ from the vocabulary i.e. $y'_{t+1} \sim f_\theta(.|\mathbf{x}, y'_{1:t})$. The environment then transitions to the next state by appending the predicted token to the current state i.e. $s_{t+1} = y'_{1:t}; y'_{t+1} = y'_{1:t+1}$. The terminal state $s_T$ is reached when either an End-of-Program token, $y'_{EOP}$ is sampled or a $y'_{Error}$ token is identified as erroneous by the compiler. Thus the final state includes complete programs and infeasible partial programs identified by the compiler. For the generate response, policy receives the reward from the reward model $\mathcal{R}$ as $G(\mathbf{x}, y'_{1:T}, \mathbf{y}) \in [-1, 1]$, where $T$ represents the MDP horizon. $\gamma$ is the discount factor and $P$ represents the transition dynamics which in our case are deterministic i.e. $P(s_{t+1}|s_t, a_t) = 1$.

### 8.4 IMPLEMENTATION DETAILS

#### 8.4.1 DATA TOKENIZATION

Table 1: Max Source and Target Sequence Lengths set to accommodate 99% of data for each model with no truncation

| DATA | MODEL | MAX SRC | MAX TRG |
|------|-------|---------|---------|
| CODENETJAVA | CODET5 | 500 | 384 |
| | CODEGEN | 500 | 384 |
| | GPT-2 | 500 | 384 |
| | GPT-NEO | 500 | 384 |
| MBJP | CODET5 | 500 | 256 |
| | CODEGEN | 500 | 256 |
| | GPT-2 | 500 | 524 |
| | GPT-NEO | 500 | 524 |
| MATHQA | CODET5 | 300 | 256 |
| | CODEGEN | 300 | 256 |
| | GPT-2 | 300 | 384 |
| | GPT-NEO | 300 | 384 |
| HUMANEVAL | CODET5 | 500 | 524 |
| | CODEGEN | 500 | 524 |
| | GPT-2 | 500 | 524 |
| | GPT-NEO | 500 | 524 |

#### 8.4.2 CODEBERT PRE-TRAINING

**Objective.** The implicit objective of the discriminator $D$ is to score the feasibility of a code sampled for a given prompt, with respect to a reference code. As the discriminator employs CodeBERT for learning representations of a code, we do contrastive pre-training of CodeBERT (Feng et al., 2020) with the aim of learning representations that are robust to semantics-preserving code transformations (Yefet et al., 2020; Jain et al., 2020). We posit that this approach will facilitate $D$ in capturing functional equivalence and appropriately score the prompt-response pairs. This in turn should encourage exploration of the program space by appropriately rewarding policies that generate programs that may not precisely match the reference programs but are functionally or semantically equivalent.

**Implementation.** To achieve our objective, we utilize a margin-based triplet loss (Weinberger & Saul, 2009) to encourage the closeness of functionally equivalent codes in the feature embedding space. For a given prompt $\mathbf{x}$, we sample an anchor code $\mathbf{y}_a$ and a positive code $\mathbf{y}_p$ (for the same prompt), as well as a negative code $\mathbf{y}_n$ (for a different prompt through negative mining). By minimizing the following objective, we aim to enhance the performance of $D$ in appropriately scoring programs:

$$\min \mathcal{L}(\mathbf{x}, \mathbf{y}_a, \mathbf{y}_p, \mathbf{y}_n) = \mathbb{E}\Big[||\text{CodeBERT}(\mathbf{x} \circ \mathbf{y}_a) - \text{CodeBERT}(\mathbf{x} \circ \mathbf{y}_p)||_2^2$$

$$-||\text{CodeBERT}(\mathbf{x} \circ \mathbf{y}_a) - \text{CodeBERT}(\mathbf{x} \circ \mathbf{y}_n)||_2^2 + \alpha\Big]$$

where, $||.||_2$ represents the $\mathcal{L}_2$ norm of a vector. `[CLS]` token of CodeBERT is used as representation for the prompt-response pair. Choice of hyper-parameters: lr= $1e-5$, num epochs= 5, margin, $\alpha = 0.5$, linear annealing with warmup= 1000 steps. We leverage CODENETJAVA (Puri et al., 2021) that has annotations for multiple functionally equivalent programs to obtain the desired data for pre-training. We point out that in the absence of data with such annotations, source-to-source semantics-preserving transformation can be used to obtain them (Wang & Christodorescu, 2019; Rabin & Alipour, 2020; Jain et al., 2020). Having such data enables us to expose CodeBERT to a diverse range of functionally equivalent code patterns, thereby enhancing its ability to capture the nuances of program equivalence.

### 8.4.3 RLCF

Table 2: Hyper-parameters for Supervised Pre-Training of $f_\theta$ on CODENETJAVA (Puri et al., 2021)

| Model | Size | $N_{XE}$ | $lr_{XE}$ | Batch Size | Gradient Accumulation |
|-------|------|----------|-----------|------------|----------------------|
| CodeGen | 350M | 5 | 5e-5 | 16 | 8 |
| CodeT5 | 220M | 5 | 5e-5 | 8 | 8 |
| CodeT5 | 770M | 5 | 5e-5 | 32 | 8 |
| GPT-2 | 124M | 5 | 5e-5 | 16 | 8 |
| GPT-2 | 1.5B | 5 | 5e-5 | 4 | 8 |

We initialise the discriminator, $D$ for each actor based on the responses sampled from it (see Step 4 in Algorithm 1), training $D$ for 5 epochs using a learning rate of 1e-5 and batch size of 32. The responses were sampled from the actor policy with a temperature of $0.6$ using top-p (nucleus) sampling with $p = 0.95$.

For each actor-LLM $f_\theta$ considered for RLCF, we used a smaller language model as Critic $v_\varphi$ such as (i) CodeT5(220M) Critic for the CodeT5(770M) Actor, (ii) GPT-2(124M) Critic for GPT-2(1.5B) Actor and (iii) CodeGen(350M) for both Actor and Critic. Specific hyper-parameter details for bootstrapping these models on CODENETJAVA (Puri et al., 2021) are provided in Table 2. We did RL training for $N_{RL} = 150K$ episodes using a batch size of 8, the learning rate of $1e-6$ and ADAMW$(0.9, 0.99, 1e-8)$ optimizer with linear decay for the learning rate. If `freezeDisc` is `False`, $D$ is updated on the sampled batch with a learning rate of $5e-7$. For PPO, we set clipping parameter $\epsilon = 0.2$. We set $\gamma = 0.99$ and $\lambda = 0.95$ in the GAE calculation. The initial KL controller coefficient $\beta$ is 0.1, which is controlled dynamically (Ziegler et al., 2019). RLCF training of CodeGen 350M and CodeT5 770M models was done on 4x16GB Tesla P100 GPUs which took approximately 5 and 7 days respectively, while training of the larger GPT2 1.5B was done 4xA6000 48GB GPUs and it took approximately 5-6 days to complete.

### 8.4.4 FINE-TUNING

Table 3: Hyper-parameters for Fine-Tuning RLCF Models and all Baselines on MBJP and MathQA

| Model | Size | $N_{epochs}$ | $lr$ | Batch Size | Gradient Acc. | Warmup Steps |
|-------|------|--------------|------|------------|---------------|--------------|
| CodeGen(multi) | 350M | 20 | 5e-5 | 8 | 8 | 200 |
| CodeT5 | 770M | 20 | 5e-5 | 8 | 8 | 200 |
| GPT-Neo | 1.3B | 20 | 5e-5 | 8 | 8 | 200 |
| GPT-2 | 1.5B | 20 | 5e-5 | 4 | 8 | 200 |
| CodeGen(multi) | 2.7B | 20 | 5e-5 | 4 | 8 | 200 |

### 8.4.5 UNBIASED ESTIMATORS FOR EVALUATION

$$metric@k := \mathbf{E}_{problems} \left[ 1 - \frac{\binom{n-c}{k}}{\binom{n}{k}} \right]$$

We use the above formulation from Chen et al. (2021a) to compute $pass@k$, $exec@k$ and $comp@k$. We sample $n > k$ programs per problem and count the number of samples $c$ (i) that compile, execute and pass all test-cases for $pass@k$, (ii) that compile and execute successfully for $exec@k$ and (iii) that compile successfully for $comp@k$. We estimate each metric using temperatures $\in \{0.2, 0.6, 0.8\}$ and choose the temperature that yields the best-performing $pass@k$ following the evaluation procedure from Chen et al. (2021a).

### 8.5 IMPLEMENTATION AND ADDITIONAL DISCUSSION: ABLATION WITH BIPOLAR-RAMP

We implemented a weakly supervised baseline trained by minimising the following token-wise Bipolar RAMP Loss (Jehl et al., 2019).

$$\mathcal{L}_{RAMP-T} = \mathbb{E}_{\left( \mathbf{x}, \mathbf{y}^+ \sim F, \mathbf{y}^- \sim f_{\theta'(n)}(.|\mathbf{x}) \right)} \Big[ \sum_{i=1}^{|\mathbf{y}^-|} \tau_i^- \log f_{\theta'(n+1)}(y_i^- | (\mathbf{x}, y_{<i}^-)$$
$$- \sum_{j=1}^{|\mathbf{y}^+|} \tau_j^+ \log f_{\theta'(n+1)}(y_j^+ | (\mathbf{x}, y_{<j}^+) \Big] \quad (3)$$

where, for given prompt $\mathbf{x}$, $\mathbf{y}^+$ is the response sampled with the prompt from $F$ and $\mathbf{y}^-$ is the response sampled from the current policy $f_{\theta'(n)}$ before the update. The token-level weights are $\tau_j^+ = 1 \ \forall j$ and $\tau_i^- = -1$ if $i$ is the location at which compilation failed else $\tau_i^- = 0$. Overall, this training also operates similarly to RLCF by discouraging erroneous programs identified by the compiler in addition to promoting the gold standard but in a supervised manner.

**Comparison with RLCF.** In our comprehensive experiments, we observed that RLCF outperforms Bipolar RAMP baseline on downstream tasks. We argue that a key factor contributing to this gap in performance is the issue of overfitting, where the Bipolar RAMP baseline succumbs to overfitting within the same number of iterations as RLCF, leading to its inferior results.

The overfitting phenomenon can be attributed to the inherent characteristics of the Bipolar RAMP model. Specifically, Bipolar RAMP adopts a strategy that increases the likelihood of the response sampled with the prompt, while simultaneously decreasing the likelihood of *only* the erroneous token identified by the compiler in the generated response. This limited perspective restricts the model's ability to generalize beyond the training data and hinders its overall performance.

On the contrary, RLCF adopts a more sophisticated approach by leveraging a denser feedback for each token generated by the actor policy. This feedback is derived from the advantages estimated by the critic network, which in turn relies on rewards or penalties provided by the grounding function, $G$ for exploring the state-action space beyond the confines of the training data. By incorporating this richer feedback signal, RLCF is capable of dynamically adjusting the likelihood of the generated response, leading to a better policy. The observed performance gap between RLCF and the Bipolar RAMP baseline on MBJP substantiates the validity of our argument.

### 8.6 IMPLEMENTATION AND ADDITIONAL DISCUSSION: ABLATION WITH CODERL

For a fair comparison with RLCF, we adapt the CodeRL (Le et al., 2022) for coarse-tuning where unit tests are not present and generated programs are solely *critiqued* based on compilation feedback.

$$\nabla_{\theta'} \mathcal{L}_{CodeRL} = -\mathbb{E}_{\left( \mathbf{x} \sim F, \mathbf{y}' \sim f_{\theta'(n)}(.|\mathbf{x}) \right)} \Big[ (r(\mathbf{y}') - r(\mathbf{y}_b)) \sum_t \hat{v}_\phi(y_t) \nabla_{\theta'} \log f_{\theta'(n+1)}(y_t | \mathbf{x}, y_{<t}) \Big] \quad (4)$$

where, for a sampled response $\mathbf{y}'$ the return $r$ is defined as $r(.) = -1$ if it gives a compilation error, $+1$ otherwise. $r(\mathbf{y}_b)$ denotes the return of a baseline response. This baseline response is greedily decoded and is utilized to mitigate variance in the gradient estimation.

Lastly, $\hat{v}_\phi(y_t)$ is the token-level intermediate return estimated by a critic network $v_\phi$ trained to infer whether the response $\mathbf{y}$ for a specific prompt $\mathbf{x}$ compiles. Similar to CodeRL (Le et al., 2022) we used transformer models of size smaller than actor as critic. In the critic architecture, the contextual hidden states for every token of the response $\{h_1, \dots h_T\}$ are max-pooled to obtain the response's representation, $h_{Pool}$. This is then fed to a Linear Layer for binary classification i.e. output $\hat{u} = \texttt{sigmoid}(\texttt{Linear}(h_{Pool}))$. Given a learned critic, the token-level return is estimated as $\hat{v}_\phi(y_t) = \texttt{sigmoid}(\texttt{Linear}(h_t))$ if the response compiles, $1 - \texttt{sigmoid}(\texttt{Linear}(h_t))$ otherwise.

**Distinctiveness of RLCF from CodeRL.** At their core, both RLCF and CodeRL aim to improve code generation, but they operate differently. In the case of CodeRL, test cases are naturally used to ensure that the policy stays on-target and generates codes that are relevant to the prompt: if the policy generates code that fails test cases, it is punished. This is where RLCF stands out. In absence of test cases, it uses a discriminator to punish generated codes that are not in-keeping with the prompt. This explains the observed superior performance of RLCF in pass@k metric (consider CODEGEN with $k = 100$, where there is an increase from 9.3% to 13.4% and CODET5 with $k = 100$, where there is an increase from 12.9% to 16.5%).

Both RLCF and CodeRL exhibit comparable comp@k results due to their integration of compiler feedback during training. However, RLCF stands out by enabling additional static analysis beyond mere compilation issues, like pinpointing unused variables. Such feedback, aimed at reducing poor coding habits (See Figure 11), is distinctive to RLCF and is not present in CodeRL.

While these findings showcase the innovations that RLCF brings, it is worth noting that RLCF and CodeRL are complementary, not competitive. Use of RLCF does not preclude the use of CodeRL. Both address different phases in the ML-for-code evolution. RLCF is meant to be used to complete pre-training on any corpus that contains compilable codes, before the model is published (which we refer to as "coarse-tuning"). CodeRL, conversely, is apt for "fine-tuning" when deploying models in test-case-available domains. In essence, CodeRL can be used for fine-tuning to further hone the model, after use of RLCF, an interesting study that we leave for future work.

### 8.7 ADDITIONAL ABLATION STUDY: IMPACT OF DESIGN COMPONENTS OF THE COMPILER FEEDBACK

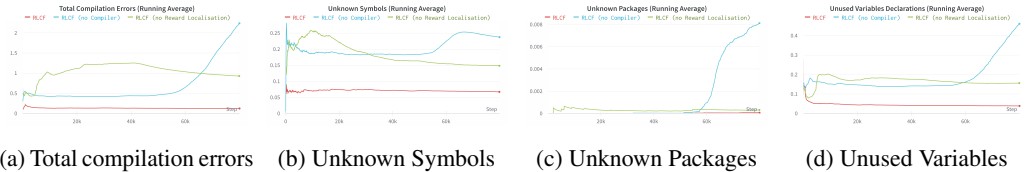

(a) Total compilation errors  (b) Unknown Symbols  (c) Unknown Packages  (d) Unused Variables

Figure 11: Plots comparing the compilation profile of response (avg. errors/issues per response) generated during on-policy coarse-tuning of CodeT5 using:- (i) proposed RLCF, (ii) RLCF with no compiler feedback and (iii) RLCF with no reward or error localization (i.e., erroneous responses are not truncated at the compiler-identified token and the reward is given at the end to the full response)

What role does the compiler play within the grounding function during model tuning? How crucial is token-level error localization and the subsequent response truncation?

To answer these, we trained `CodeT5` using RLCF but ablated each component. (i) No compiler signal: We observed a significant divergence in learned policy from its initial distribution when compiler feedback is absent during on-policy training (refer to Figure 11), leading to a sharp surge in compilation errors by the resulting model. This underscores the necessity of compiler feedback in maintaining the correct code syntax and semantics. (ii) No error localisation: Without this feature, the LLM's on-policy training produces substantially more compilation errors per generated response, evidently shown by the 'RLCF with-no-localization' variant. Localization helps in pinpointing and curbing specific errors and undesired coding patterns, like the declaration of unused variables, as

shown in Figure 11d. Thus, our results indicate that precise error detection and localization in code provide a more effective credit allocation for reinforcement learning.

## 8.8 ADDITIONAL ABLATION STUDY: ADVANTAGE OF DOING RLCF WITH A PRE-TRAINED DISCRIMINATOR

Table 4: Additional discriminator ablation study with CodeT5(770M) on MBJP Test

| MODEL | PASS@K | | | EXEC@K | | | COMP@K | | |
|---|---|---|---|---|---|---|---|---|---|
| | K=1 | K=10 | K=100 | K=1 | K=10 | K=100 | K=1 | K=10 | K=100 |
| CODET5 (770M) | 4.17 | 7.66 | 9.27 | 48.85 | 79.52 | 88.65 | 56.44 | 84.91 | 91.23 |
| + RLCF- PRETRAIN | 5.18 | 9.57 | 13.40 | 53.80 | 80.44 | 88.65 | **63.84** | 86.44 | 91.75 |
| + RLCF | **6.32** | **12.26** | **16.49** | **54.09** | **82.52** | **89.69** | 62.68 | **87.56** | **92.78** |

In the Table 4, we present the benefits of pre-training the discriminator $D$ for the grounding function $G$ described in the paper on responses sampled from the bootstrapped policy, as opposed to one that is learned from scratch with the policy. While both options result in better performance than the baseline, the RLCF policy starting with the pre-trained $D$ achieves an additional improvement in performance across all metrics, particularly in pass@k metrics. We argue that this can be attributed to the pre-trained $D$ network's ability to enter RL-training with the knowledge of how to score responses for a given prompt and compare them. As a result, the policy during RLCF must better understand the prompt to generate responses that fool the discriminator, while the untrained discriminator must first learn how to score programs and should be relatively easier to fool.

## 8.9 ADDITIONAL BENCHMARK EVALUATION: PERFORMANCE EVALUATION ON HUMANEVAL FOR JAVA

Table 5: We zero-shot test baselines against models coarse-tuned with RLCF on HumanEvalJava (Athiwaratkun et al., 2022)

| MODEL | PASS@K | | EXEC@K | | COMP@K | |
|---|---|---|---|---|---|---|
| | K=10 | K=100 | K=10 | K=100 | K=10 | K=100 |
| CODEGEN (350M) | 9.94 | 11.18 | 90.83 | 97.51 | 93.72 | **99.37** |
| + RLCF | **11.50** | **13.66** | **93.17** | **98.13** | **94.89** | 98.19 |
| CODET5 (770M) | 7.94 | 11.18 | 80.62 | 91.30 | 84.72 | 93.78 |
| + RLCF | **9.97** | **11.80** | **84.68** | **93.78** | **87.87** | **96.27** |
| GPT2 (1.5B) | 5.63 | 7.45 | 79.53 | 88.19 | 81.63 | 89.44 |
| + RLCF | **8.03** | **13.66** | **86.03** | **96.89** | **88.05** | **96.89** |

In Table 5, we show performance comparison on 161 problems HumanEval for Java, where we can again observe from modest (CodeGen, CodeT5) to significant improvements (GPT-2) achieved by RLCF coarse-tuned models.

## 8.10 ILLUSTRATION: CODENETJAVA DATA PREPARATION

(a) Full Program from CODENETJAVA

```java
import java.util.Scanner;
import java.util.Arrays;

public class Main{
    public static void main(String[] args){
        Scanner scan = new Scanner(System.in);
        int[] heights = new int[10];
        for(int i = 0; i < 10; i++){
            heights[i] = scan.nextInt();
        }

        Arrays.sort(heights);
        for(int i = 9; i >= 7; i--){
            System.out.println(heights[i]);
        }
    }
}
```

(b) Training Sample 1

```java
import java.util.Scanner;
import java.util.Arrays;

public class Main{
    public static void main(String[] args){
        /*<your_code_here>*/
}
```

(c) Training Sample 2

```java
import java.util.Scanner;
import java.util.Arrays;

public class Main{
    public static void main(String[] args){
        Scanner scan = new Scanner(System.in);
        int[] heights = new int[10];
        for(int i = 0; i < 10; i++){
            heights[i] = scan.nextInt();
        }
        /*<your_code_here>*/
}
```

Figure 12: Demonstration of training sample preparation from CODENETJAVA (Puri et al., 2021) for RLCF. A random instruction within a method body is selected, and the subsequent content is masked. The model's task is to complete the method.

## 8.11 ILLUSTRATION: COMPILER FEEDBACK LED RESPONSE TRUNCATION WITHIN THE ROLLOUTTRAJECTORY ROUTINE

(a) Prompt consisting of NL Description and Historical Code

```
1  /**
2  Takahashi went to an all-you-can-eat buffet with N kinds of dishes and
       ate all of them (Dish 1, Dish 2, ..., Dish N) once. The i-th dish (1
       <= i <= N) he ate was Dish A_i. When he eats Dish i (1 <= i <= N), he
        gains B_i satisfaction points. Additionally, when he eats Dish i+1
       just after eating Dish i (1 <= i <= N - 1), he gains C_i more
       satisfaction points. Find the sum of the satisfaction points he
       gained. Standard Input in the following format: N A_1 A_2 ... A_N B_1
        B_2 ... B_N C_1 C_2 ... C_{N-1}. Print the sum of the satisfaction
       points Takahashi gained, as an integer.
3  */
4
5  public static void main(String[] args) throws IOException {
6      Scanner scan = new Scanner(System.in);
7      int N = scan.nextInt();
8      int sum=0;
9      /*<your_code_here>*/
```

(b) Response from CodeT5-770M. Compiler throwing *Error*: Cannot find symbol **B** and **a** (Highlighted in Red)

```
1      int[] A = new int[N];
2      for(int i=0;i<N;i++){
3          A[i] = scan.nextInt();
4      }
5      for(int i=0;i<N;i++){
6          int b = scan.nextInt();
7          sum += B[a[i]];
8      }
9      for(int i=0;i<N-1;i++){
10         int c = scan.nextInt();
11         if(A[i]-1==a[i+1]-1){
12             sum += c;
13         }
14     }
15     System.out.println(sum);
16     scan.close();
17 }
```

(c) Response truncated at the first erroneous token

```
1      int[] A = new int[N];
2      for(int i=0;i<N;i++){
3          A[i] = scan.nextInt();
4      }
5      for(int i=0;i<N;i++){
6          int b = scan.nextInt();
7          sum += B
```

Figure 13: Illustrating how response is truncated in the ROLLOUTTRAJECTORY routine

In Figure 13, we demonstrate how the grounding function limits the generated response to the first detected error. For example, if the compiler raises a *Symbol Not Found* error for undeclared or uninitialized variables at locations marked in red, the ROLLOUTTRAJECTORY identifies the first of these errors based on its position in the code, localizes it at the token level, truncates the response, and the grounding function returns the truncated response with a $-1$ reward.

### 8.12 ILLUSTRATION: DISCRIMINATOR SCORING COMPARISON ON RESPONSES PASSING ALL STATIC CHECKS

In Figure 14, we demonstrate how the discriminator LLM identifies and penalizes nonsensical responses, such as the one generated by CodeT5 during RLCF as shown in Figure 14d. We also showcase some equally impressive responses, as shown in Figure 14c, that are rewarded accordingly.

It is important to note that the generated response in Figure 14c may not precisely match the ground-truth response in every aspect. For instance, the shown response accepts inputs differently and tests the Pythagorean theorem as $c^2 = a^2 + b^2$ instead of $a^2 + b^2 = c^2$ (different order). This difference would have been otherwise penalized by supervised learning-based baselines. This highlights how RLCF encourages exploration of the program state-space while maintaining compliance with static checks and is rewarded equally.

(a) Prompt consisting of NL Description and Historical Code

```
/**
Is it a Right Triangle? Write a program which judges wheather given
    length of three side form a right triangle. Print "YES" if the given
    sides (integers) form a right triangle, "NO" if not so. Input
    consists of several data sets. In the first line, the number of data
    set, N is given. Then, N lines follow, each line corresponds to a
    data set. A data set consists of three integers separated by a single
     space. For each data set, print "YES" or "NO".
*/

public static void main(String[] args) throws IOException {
    Scanner sc = new Scanner(System.in);
    int n = sc.nextInt();
    for (int i = 0; i < n; i++){
        /*<your_code_here>*/
```

(b) Ground truth response

```
        int a[] = new int[3];
        for (int j = 0; j < 3; j++){
            a[j] = sc.nextInt();
        }
        Arrays.sort(a);
        if (a[0]*a[0] + a[1]*a[1] == a[2]*a[2]){
            System.out.println("YES");
        }else{
            System.out.println("NO");
        }
    }
}
```

(c) Response by CodeGen-350M receiving High Score (+1) from Discriminator

```
1          int a[] = new int[3];
2          a[0] = sc.nextInt();
3          a[1] = sc.nextInt();
4          a[2] = sc.nextInt();
5          Arrays.sort(a);
6          if (a[2] * a[2] == a[1] * a[1] + a[0] * a[0]){
7              System.out.println("YES");
8          } else {
9              System.out.println("NO");
10         }
11     }
12     sc.close();
13 }
```

(d) Response by CodeT5-770M receiving Low Score (<0) from Discriminator

```
1          int[] a = new int[3];
2          for(int j=0; j<3; j++){
3              a[j] = sc.nextInt();
4          }
5          Arrays.sort(a);
6          int x = a[0] + a[1];
7          int y = a[2];
8          int z = a[0];
9          if(x*x + y*y == x*z){
10         }else if(x*x + y*y == x*y+x*y){
11             if(x*x + y*y == x*y+x*z){
12                 if(x*x + y*y == x*y){
13                     if(x*x + y*y == x*y){
14                         ir = true;
15                     }
16                 }else {
17                     ir = false;
18                 }
19             }
20         }
21         if(ir){ System.out.println("YES");
22         } else {
23             System.out.println("NO");
24     }
25 }
```

Figure 14: Illustrating how Discriminator scores compilable codes generated by different models

