# OpenReview forum: "Coarse-Tuning Models of Code with Reinforcement Learning Feedback"
_ICLR.cc/2024/Conference — Submitted to ICLR 2024_

### Official Review · Reviewer_HG9j · 2023-10-31

**Soundness:** 2 fair
**Presentation:** 2 fair
**Contribution:** 2 fair
**Rating:** 5
**Confidence:** 3

**Summary:**

The paper proposes tuning language models (pre-trained on code) for a specific programming language by leveraging static analysis tools (like those used in compilers). The static analysis tools are combined with a separate language model (CodeBERT) to craft a reward function. This reward is used with PPO to train the language model. Experiments with three different language models (up to 1.5B parameters) on code examples in Java demonstrate that the proposed method is able to improve the performance over the pre-trained model.

**Strengths:**

Using the compiler (and related tools) to improve the ability of language models to generate code is a promising direction. This approach reflects how human programmers also require feedback from the programming environment to hone their skills, whereas solely reading code is not sufficient.

Many commonly used programming languages are supported with static analysis tools, so the proposed method appears to be quite general.

The presentation of the method and the results is clear.

**Weaknesses:**

The experiments do not seem to control for the amount of compute. Comparing RLCF to a baseline that uses 0 compute (after pre-training) does not provide a useful comparison. A more informative baseline would be one that uses the same amount of compute as RLCF but uses the standard LM loss function instead. The "+Mono" baseline in Table 7 goes in this direction, but it does not apply the same amount of compute as RLCF.

The proposed method requires a so-called discriminator D that is part of the grounding function. This discriminator is a separate LM (not the same as the one that is being tuned to generate better code). The specific model used here is a pre-trained CodeBERT. It is unclear why this specific choice was made and how this choice impacts the results. While it appears that the pre-training here isn't strictly necessary for RLCF to yield improvements (based on Table 4 in the appendix), it does look like pre-training has a very significant effect. But using a pre-trained model puts into question whether performance improvement primarily arises due to distillation effects? Also, how well does CodeBERT perform on the tasks? When using a different model as the discriminator, do the numbers still look the same?

The results seem to suggest that RLCF is effective in improving the rate at which the LM samples compilable and executable programs. However, this improvement does not appear to correlate as much as expected with the improvement in the proportion of samples that pass the test cases. Currently it is unclear to me how this improvement in passed test cases arises and I'm more inclined to believe that the majority of it is due to distilling "Java code knowledge" from CodeBERT. I'd suggest running a RLHF baseline with the HF replaced by CodeBert.

**Questions:**

Are the model descriptions for pre-trained and not pre-trained in Table 4 in the appendix mixed up?

See weaknesses for more questions.

---

> ### Author Response · Authors · 2023-11-15
>
> - ***“The experiments do not seem to control for the amount of compute … but it does not apply the same amount of compute as RLCF.”***: We interpret the reviewer's concern about "compute" as referring to the amount of resources consumed, including the number of samples, batch size, and training iterations. To ensure a fair comparison, our coarse-tuning baselines (+Mono, +BiPolarRamp, +CodeRL) and RLCF are trained with identical hyperparameters for a given model size. This includes an equal number of samples from CodeNetJava, matching batch sizes, and total training iterations. The variations lie only in the objective function being optimized. The baseline that does not undergo coarse-tuning can be viewed as utilizing 0 compute. However it is outperformed by RLCF across all metrics for all models with just few rounds of training (see Figure 3). This underscores its efficacy as a cost-effective means of enhancing model performance for downstream tasks.
>
>
> - ***“The proposed method requires a so-called discriminator D that is part of the grounding function …”***: The choice of CodeBERT for the discriminator in the grounding function is based on its suitability for code-understanding tasks. While alternatives like GraphCodeBERT exist, we emphasize that RLCF is a versatile framework where compilers and discriminators can be interchanged for feedback. The use of a pre-trained discriminator, while not strictly necessary for RLCF improvements (as indicated in Table 4 in the appendix), does have a significant impact. As discussed in Section 8.8, this can be attributed to the pre-trained D network’s ability to enter RL-training with the knowledge of how to score responses for a given prompt and compare them. Exploring discriminators that are constructed from other LLMs cooperating with advanced static-analysis tools is an avenue for future research.
>
>
> - ***“The results seem to suggest that … distilling "Java code knowledge" from CodeBERT.”***: The observed discrepancy between the improvement in Comp@k/Exec@k accuracies and Pass@k is anticipated. Higher improvement in compilation accuracy stems from the compiler feedback that precisely localizes issues in generated responses, while pass@k relies on how well the policy has learned to understand the prompt, which in the absence of test cases to indicate functional correctness has to be estimated via some Grounding function.
>
>     The improvements in pass@k are a result of the grounding function's design in RLCF, where the policy engages in an adversarial competition with the Discriminator. This process aims to generate responses that are  equally plausible compared to ground-truth responses, and possibly more plausible. It is essential to note that such an effect goes beyond a simple distillation of CodeBERT's knowledge of Java code; instead, CodeBERT is required to understand how well the generated Java code aligns with the given prompt.
>
>
> - ***“I'd suggest running a RLHF baseline with the HF replaced by CodeBert.”***: The coordinated feedback from Compiler + Discriminator in RLCF can be indeed interpreted as replacing the Human Feedback component of RLHF. We have discussed such a comparison between RLCF and RLHF in the Related Work section. For a baseline similar to RLHF that only uses CodeBERT for feedback, we ablated the compiler signal and observed a significant divergence in learned policy from its initial distribution. Refer to Section 8.7 in the Appendix for more details and discussion.
>
>
> - ***“Are the model descriptions for pre-trained and not pre-trained in Table 4 in the appendix mixed up?”***: The terms "pretrain" and "- pretrain" in Table 4 refer to whether the Discriminator enters RLCF tuning with or without learning the Grounding, as clarified in the discussion in Section 8.8. If the terminology is unclear or confusing, we will use different terms for clarity in the context of this ablation study.

---

> ### Comment · Reviewer_HG9j · 2023-11-23
>
> Thanks to the authors for the response.
>
> > The terms "pretrain" and "- pretrain" in Table 4 refer to whether the Discriminator enters RLCF tuning with or without learning the Grounding
>
> I see, so the - pretrain refers to "subtracting" the pretraining aspect? Normally one would read it as a hyphen.
>
> > As discussed in Section 8.8, this can be attributed to the pre-trained D network’s ability to enter RL-training with the knowledge of how to score responses for a given prompt and compare them.
>
> Since a substantial part of the overall improvement of RLCF comes from using a pre-trained D I would expect more analysis and discussion around it. For example how good is it at discerning correct from incorrect programs?
> The lack of analysis around the discriminator makes it difficult to follow why "... such an effect goes beyond a simple distillation of CodeBERT's knowledge of Java code".
>
> Given that my concerns remain insufficiently addressed, I stick to my initial score.

---

### Official Review · Reviewer_1rNz · 2023-11-01

**Soundness:** 3 good
**Presentation:** 4 excellent
**Contribution:** 2 fair
**Rating:** 3
**Confidence:** 3

**Summary:**

This work propose a method to fine-tune an LLM for code generation using RL. The feedback consists of two components: localized compiler errors to encourage the LLM to generate code which compiles, and a CodeBERT-based discriminator which tries to distinguish between the code generated by the LLM vs. the ground-truth solution (conditioned on the prompt). Results show improvements on several baseline code generation models (up to 1.5B parameters) for Java code generation.

**Strengths:**

The idea to fine-tune an LLM for code generation using only feedback from static analysis only is, as far as I can tell, novel. The writing is also clear and well motivated. As code generation is a major application of LLMs, the work has potential for significant impact as well. The experimental results are also relatively comprehensive, including a slate of baselines for comparison.

**Weaknesses:**

Prior works (see below) have used a combination of static and dynamic analysis as an RL reward for fine-tuning LLMs for code generation, which limits the novelty.

The decision to focus on Java, motivated by the authors for its static typing and availability of static analyzers, makes comparisons with existing works difficult, which have almost universally adopted Python as the language of choice. The benchmark datasets (MathQA and MBJP) were originally written in Python, and then transpiled automatically into Java. Even the dataset used in this work for finetuning, CodeNetJava, has a Python equivalent. Additionally, this work leverages feedback from the static analyzer consisting solely of the location of the compiler error, which should be available for Python as well.

For instance, RLTF [1], which is another RL for code generation technique, achieves 30.4 pass@1 on MBPP (the original python dataset) using the 770M CodeT5 as the base model. This compares with 6.6% for RLCF (this work) on MBJP (using the same base model, which starts around the same pass@1 of ~4% for MBPP).

Ideally, I would have preferred the experiments to have been done in Python, but at the very least, the related work should include a discussion of works which apply RL for code generation. For instance CodeRL [2] is mentioned as a baseline, but there is no comparison in the related work. As far as I can tell, the specific design (using a discriminator as well as returning the location of a compile error) is novel, but there are certainly parallels to prior work (for instance, CodeRL uses a critic to return localized information). Another highly relevant work is [3], which combines RL with an AST-based syntactic match metric as well as a dataflow graph based semantic match metric (both of which are static rather than dynamic analyses).

Finally, I'm not sure the ablation study for CodeRL is a fair comparison, as CodeRL includes a number of other components beyond simply an RL reward for compile errors (such as the aforementioned critic network). This can be addressed by renaming the ablation to something other than CodeRL.

[1] Liu, Jiate, et al. "RLTF: Reinforcement Learning from Unit Test Feedback." arXiv preprint arXiv:2307.04349 (2023).
[2] Le, Hung, et al. "Coderl: Mastering code generation through pretrained models and deep reinforcement learning." Advances in Neural Information Processing Systems 35 (2022): 21314-21328.
[3] Shojaee, Parshin, et al. "Execution-based code generation using deep reinforcement learning." arXiv preprint arXiv:2301.13816 (2023).

**Questions:**

Can you provide a comparison of your methods with the 3 works cited above?

Can you ablate the localization aspect of the compiler feedback? i.e., just return -1 for compile errors (while maintaining the discriminator).

It would also be good to swap out the learned discriminator with the DFG-based metric from [3] above.

---

> ### Author Response · Authors · 2023-11-15
>
> - ***“RLTF [1], which is another RL-for-code-generation technique, achieves … ~4% for MBPP)”***: Comparing RLTF's reported results on MBPP with RLCF's results on MBJP might be misleading. RLCF operates in a coarse-tuning setting without access to test cases, while RLTF's reported results on MBPP involve fine-tuning with unit-test feedback. Additionally, RLCF addresses a different aspect of the ML-for-Code cycle, leveraging large-scale datasets used for pre-training where unit tests are unavailable. CodeRL, PPOCoder, and RLTF, on the other hand, are designed for fine-tuning with available test cases.
>
>
> - ***“For instance CodeRL [2] … (for instance, CodeRL uses a critic to return localized information). Finally, I'm not sure the ablation study for CodeRL is a fair comparison … other than CodeRL”*** : We appreciate the reviewer's suggestion for experiments in Python, which we will consider for future work. Our Related Work section already includes a comparison with CodeRL (see last paragraph of Section 2). Regarding the ablation study with respect to CodeRL, our implementation of CodeRL follows its specified architecture where we indeed incorporate the critic network. We only removed the reward components dependent on unit-test feedback for a fair comparison with RLCF. Detailed information on the CodeRL implementation and comparison is available in Section 8.6 of the Appendix.
>
>
> - ***“Can you provide a comparison of your methods with the 3 works cited above? It would also be good to swap out the learned discriminator with the DFG-based metric from [3] above.”***: We have included a comprehensive comparison with CodeRL in the paper. Regarding RLTF and PPOCoder, we emphasize their lack of a grounding function for strict alignment to prompts. While PPOCoder utilizes AST and DFG-based matching between generated and ground-truth responses, we argue that functionally equivalent programs with different structural compositions may receive poor AST and DFG-based matching scores. Large language models (LLMs) like CodeBERT, which learn representations from a diverse set of code, are adept at handling such structural variations and possess contextual understanding of responses with respect to prompts. This enables them to recognize similarities that might be overlooked by DFG matching, which often focuses on local relationships. We will incorporate this discussion of RLTF [1] and PPOCoder [3] in the final manuscript.
>
>
> - ***“Can you ablate the localization aspect of the compiler feedback? i.e., just return -1 for compile errors (while maintaining the discriminator).”***: Ablating the localization aspect of compiler feedback by just returning -1 for compile errors, while maintaining the discriminator, resulted in significantly more compilation errors per generated response. For a detailed discussion on this ablation, we encourage the reviewer to refer to Section 8.7 of the manuscript.

---

> > ### Comment · Reviewer_1rNz · 2023-11-21
> >
> > > our implementation of CodeRL follows its specified architecture
> >
> > Thanks for the clarification, this addresses my concern on the comparison to CodeRL.
> >
> > > we argue that functionally equivalent programs...
> >
> > I would rather see this claim supported by empirical evidence.
> >
> > In general my concerns regarding prior work remain largely unaddressed, given that this paper considers a restricted setting without unit tests, and RLTF and PPOCoder both already have substantial portions dedicated to compiler feedback.
> >
> > >  Ablating the localization aspect of compiler feedback by just returning -1 for compile errors, while maintaining the discriminator, resulted in significantly more compilation errors per generated response.
> >
> > I took a look at Section 8.7 and am having a hard time following the plots. Should b, c, and d sum up to a? Also, is there a plot showing the impact on functional correctness (rather than compiler errors)?

---

> > > ### Author Response · Authors · 2023-11-21
> > >
> > > We appreciate the reviewer's concern and would like to provide further clarification:
> > >
> > > 1. **I would rather see this claim supported by empirical evidence...:**
> > >    Our claim regarding higher accuracy in recognising functionally equivalent programs is supported by the performance of large language models (LLMs) like CodeBERT on benchmarks such as BigCloneBench (See CodeXGlue '21). CodeBERT achieved the highest F1 score among pure graph-based matching algorithms.
> > >
> > > 2. **Comparison with RLTF and PPOCoder:**
> > > In contrast to CodeRL, RLTF, and PPOCoder, which rely on the presence of test cases for functional grounding or prompt alignment, we contend that this assumption makes their setting more restrictive. Unit tests are frequently absent from large-scale pre-training datasets used for training language models. Our demonstration with CodeRL illustrates that when these baselines are cast in a more general setting where unit tests are lacking, relying solely on compiler feedback, they underperform.  RLCF, utilizing coordinated feedback from the compiler and "grounding function", emerges as a more suitable choice in scenarios where unit tests are not available.
> > >
> > > 3. **Plots in Section 8.7:**
> > >    The plots in Section 8.7 illustrate average error rates while training CodeT5 model, with (b), (c), and (d) highlighting specific error categories that do not sum upto (a). Plot (a) presents total compilation errors across all categories, encompassing other issues like bad operand types, incompatible types, or type mismatches flagged by the compiler. The selection of (b)-(d) categories was handpicked to showcase the trends observed in this ablation study, reflecting similar patterns in other error categories.
> > >
> > > 4. **Functional Correctness Evaluation:**
> > >    Due to the absence of test cases, measuring functional correctness during RLCF tuning is challenging. Thus we are only able to observe pass@k on fine-tuning datasets (Fig. 3), where unit tests are available. There the higher pass@k suggests that our grounding function enhances the model's ability to understand natural language prompts and generate functionally correct programs.

---

### Official Review · Reviewer_t6g5 · 2023-11-01

**Soundness:** 3 good
**Presentation:** 3 good
**Contribution:** 2 fair
**Rating:** 6
**Confidence:** 4

**Summary:**

This work proposes a new approach to to program synthesis using reinforcement learning and feedback from a grounding function. The experiments show promising results in improving the performance of LLM-generated programs.

**Strengths:**

+ The RLCF proposed in this work uses feedback from both compiler-derived feedback and LLM feedback.

+ The proposed approach is model- and language-agnostic, making it possibly applicable to various programming languages and models.

+ This work presents empirical evaluations on the MBJP and MathQA tasks for Java, showing promising results.

**Weaknesses:**

- The proposed approach is limited to larger dataset due to the fact that CODENETJAVA does not consider dependencies on user-defined packages or libraries.

- The paper only evaluates the proposed approach on two specific tasks for Java, which may not be representative of other programming languages or models.

- This work does not provide a comparison of the proposed approach with existing state-of-the-art LLMs like GPT3 or GPT4.

- How this approach can work with existing pre-trained code-specific LLMs is missing.

**Questions:**

Please check the Weaknesses for detailed questions to be answered.
- Is the proposed evaluation and experiments representative enough for other programming languages or models?
- How can this approach work with existing pre-trained code-specific LLMs is missing?
- Is it possible fro authors to compare the proposed method with existing state-of-the-art LLMs like GPT3 or GPT4?

---

> ### Author Response · Authors · 2023-11-15
>
> - ***“The proposed approach is limited to larger dataset due to the fact that CODENETJAVA does not consider dependencies on user-defined packages or libraries.”***: The use of CodeNetJava, which excludes user-defined packages, is a deliberate choice to streamline static analysis during training and facilitate rapid prototyping of RLCF. This dataset choice doesn't limit the applicability of RLCF, as the framework is equally applicable on larger datasets containing user-defined packages. For such datasets, incremental compiler tools like Bazel can be leveraged as they efficiently rebuild modified program parts considered during RLCF-based model training.
>
> - ***“Is the proposed evaluation and experiments representative enough for other programming languages or models?”***: The RLCF framework is designed to be applicable to all languages and models of various sizes. While our experiments focus on Java and a range of model sizes (from 350M to 1.5B), the extension of RLCF to other languages and models is a straightforward task, and we acknowledge this as a potential avenue for future work.
>
> - ***“How can this approach work with existing pre-trained code-specific LLMs is missing?”***: The selected CodeGen (Decoder only) and CodeT5 (Encoder-Decoder) models are code-specific language models (LLMs), and our study demonstrates improvements in generated code with RLCF applied to these models. The RLCF framework is adaptable and can be similarly applied to other existing code-specific LLMs.
>
> - ***“Is it possible fro authors to compare the proposed method with existing state-of-the-art LLMs like GPT3 or GPT4?”***: GPT models, notably GPT-3 and GPT-4, are significantly larger than the models examined in this paper. Unfortunately, their usage comes with associated computational costs, which proved difficult to procure in an academic setting. We can only ask that the reviewer evaluate our proposal on the models that we were able to run.

---

### Official Review · Reviewer_8pcA · 2023-11-02

**Soundness:** 3 good
**Presentation:** 3 good
**Contribution:** 2 fair
**Rating:** 5
**Confidence:** 4

**Summary:**

The paper introduces Reinforcement Learning with Coordinated Feedback (RLCF), a new approach to enhancing the capabilities of LLMs in program synthesis. Traditional next-token prediction training objective overlooks the syntax and semantic constraints in code. RLCF aims to address this by retraining LLMs using RL, incorporating feedback from a compiler, and a separate LM that compares generated code against a reference. This "coarse-tuning" process occurs after initial pre-training but before task-specific fine-tuning. RLCF's effectiveness is demonstrated in Java datasets (MBJP and MathQA), showing that it significantly improves the probability of generating correct, compilable, and executable code.

**Strengths:**

A novel attempt to integrating compiler feedback in RL based code generation.

Paper demonstrates an innovative approach in using a hybrid grounding function that incorporates both a compiler and a discriminator LLM. This enhances the reliability and relevance of the generated code, making the system robust against producing syntactically correct but contextually irrelevant responses.

**Weaknesses:**

Compiler limitations: The use of a compiler in the grounding function inherently relies on the limitations and capabilities of the chosen compiler. Different compilers might have varying levels of strictness or support for language features, potentially leading to inconsistencies in how code is evaluated. This could result in a situation where the model generates code that is deemed correct by one compiler but not by others.

From the perspective of practical implementation in developer tools, the requirement to utilize both a compiler and an additional discriminator model could present significant challenges.

There is a need for more robust RL baselines, such as the RLHF (for instance, with binary reward)

**Questions:**

The study compares the base pre-trained CodeGen model with the version enhanced by RLCF. Would it be feasible to include a comparison with a supervised fine-tuned variant of the CodeGen? For instance, we could create a collection of "gold standard" examples that both compile successfully and are preferred by the discriminator for SFT. This could separate the improvement from the coarse tuning technique itself vs better quality data.

The process of generalizing this approach across various programming languages, and different compilers might represent a challenge. If possible, add more tests or discuss it.

Additionally, a comparison with state-of-the-art reinforcement learning techniques that use feedback would be valuable. Utilizing a compiler to generate binary rewards could help assemble a training dataset suitable for a conventional RLHF or RLAIF frameworks, providing a stronger baseline to evaluate the effectiveness of RLCF.

---

> ### Author Response · Authors · 2023-11-15
>
> - ***"Compiler Limitations ... deemed correct by one compiler but not by others."***: We agree that different compilers might enforce different syntactic and semantic checks and could lead to differing analyses of dynamically typed languages like Python. For strongly typed languages like Java considered for RLCF, compilers are all well-tested and quite faithful to the language standard. This would typically not lead to inconsistencies.
>
> - ***"From the perspective of practical implementation in developer tools the requirement to utilize both a compiler and an additional discriminator model could present significant challenges."***: The discriminator and compiler are solely utilized in the model-training phase. In practice during deployment in developer tools, only the RLCF-tuned model would be used, mirroring the deployment setup of tools like Codex.
>
> - ***"There is a need for more robust RL baselines such as the RLHF"***: Implementing RLHF, particularly with a binary reward, demands expert human judgment for model-generated samples, posing significant challenges in the context of code. We address these limitations in the Related Work section.
>
> - ***"The study compares the base pre-trained CodeGen … vs better quality data"***: We've implemented a 'Mono' Baseline, a supervised variant trained on gold-standard responses from the dataset. This baseline uses samples that successfully compile and align with prompts, providing a comparison to gauge the impact of the RLCF technique on top of high-quality data.
>
> - ***"The process of generalizing this approach across various programming languages, and different compilers might represent a challenge. If possible, add more tests or discuss it."***: The RLCF framework is designed for strongly typed languages with standard compilers, such as Java. Extending it to dynamically typed languages like Python is more challenging as it requires robust static analysis tools. This is why we prototyped the method using Java.
>
> - ***"Additionally, a comparison with state-of-the-art reinforcement learning techniques … effectiveness of RLCF."***: We have compared RLCF against CodeRL, a strong RL baseline, showcasing improvements with RLCF. CodeRL itself relies on a dataset assembled with binary rewards from the compiler. Further details are available in Appendix Sec 8.6 for reference.

---

### Meta-Review · Area_Chair_MszC · 2023-12-13

**Metareview:**

The paper introduces Reinforcement Learning with Coordinated Feedback (RLCF), an approach to improving LLMs' capability in program synthesis. It employs reinforcement learning, incorporating feedback from a compiler and a language model. While the work is commended for its innovative use of a hybrid grounding (as feedback), concerns are raised about potential inconsistencies due to compiler limitations and the practical challenges of implementing both components in developer tools. Other concerns include its limited novelty considering similar studies like CodeRL and RLTF, and failing to connect to these approaches at a higher level.

**Justification For Why Not Higher Score:**

NA

**Justification For Why Not Lower Score:**

NA

---

### Decision · Program_Chairs · 2024-01-16

Reject